# Assessment of Diet Quality and Adherence to Dietary Guidelines in Gastrointestinal Cancer Survivors: A Cross-Sectional Study

**DOI:** 10.3390/nu12082232

**Published:** 2020-07-27

**Authors:** Sara Moazzen, Francisco O. Cortés-Ibañez, Barbara L. van Leeuwen, Behrooz Z. Alizadeh, Geertruida H. de Bock

**Affiliations:** 1Department of Epidemiology, University Medical Center Groningen, University of Groningen, 9700RB Groningen, The Netherlands; s.moazzen@umcg.nl (S.M.); f.o.cortes.ibanez@umcg.nl (F.O.C.-I.); b.z.alizadeh@umcg.nl (B.Z.A.); 2Departments of Surgical Oncology, University Medical Center Groningen, University of Groningen, 9700RB Groningen, The Netherlands; b.l.van.leeuwen@umcg.nl

**Keywords:** gastrointestinal neoplasms, esophageal neoplasms, stomach neoplasms, colorectal neoplasms, cancer survivors, diet

## Abstract

Diet quality among short- and long-term gastrointestinal (GI) cancer survivors with different tumor sites was investigated compared to a reference population cohort. Diet quality of GI cancer survivors (*n* = 307) was compared to an age- and sex-matched reference population with no history of cancer (*n* = 3070). All were selected from Lifelines, a population-based cohort. GI cancers were defined as having a history of cancer of the bowel, esophagus, or stomach. Diet quality was assessed by a self-administrated food frequency questionnaire in terms of: (i) Lifelines Diet (LLD) scores, where higher scores indicate higher diet quality; (ii) the adherence to dietary guidelines, quantified by the percentage of meeting dietary recommendations, as given by Dutch dietary guidelines; and (iii) the mean daily intake of food components. All analyses were adjusted for lifestyle factors. Diet scores in GI cancer survivors were not different from the reference population (OR = 0.97, 95% CI: 0.73–1.23). Stratification for time since diagnosis and tumor site gave similar results. The intake of vegetables, unsweetened dairies, and nuts and legumes was almost 50% lower than the recommended amount, and the mean intake of unhealthy food components was at least one serving/day among GI cancer survivors, as well as in the reference population. In the long run, GI cancer survivors do not differ from the reference population in their diet quality. In conclusion, both groups can improve their diet quality.

## 1. Introduction

Gastrointestinal (GI) cancers, including bowel, gastric, and esophagus cancers, are among the most common malignant diseases worldwide, affecting over 3 million new individuals annually.

In recent years, as a result of improvements in early diagnosis and the efficiency of therapies, there has been a constant increase in GI cancer survival rates in European countries [1]. A diagnosis of GI cancer may have a large impact on a person’s lifestyle, and especially on diet. Difficulties in food ingestion/digestion following surgical treatment or routine chemo/radio therapies, as well as food restrictions, specifically in patients with esophagus and gastric cancers, lead to the possibility of an undesired diet quality among GI cancer survivors [2,3]. It is also known that a healthy diet will improve outcomes after a diagnosis of GI cancer [4].

Recent evidence indicates a need among GI cancer survivors for information on diet and nutrition to improve diet quality [3]. Nonetheless, related investigations yield inconsistent findings; while some studies demonstrate improved dietary awareness [5], or a stable adherence to healthy diets [6], other studies demonstrated a poor diet quality status in cancer survivors [2,3,7]. This inconsistency might be partly due to the variety of study populations and the associated cancer types, cultural differences, variation in time since diagnosis, inconsistent methods of dietary assessment, or methods of dietary assessments which are not in accordance with evidence-based dietary guidelines [8]. In addition, the studies published thus far scarcely give a clear overview of the food components of the evaluated diets. For that reason, there is a need to further characterize the diet quality among GI cancer survivors [9,10,11,12].

Since the evidence on the state of diet quality in GI cancer survivors is inconclusive, we evaluated the diet quality among GI cancer survivors with different times since diagnosis and different tumor sites, and compared this with an age- and sex-matched cohort of individuals who had no history of cancer. In addition, we investigated the adherence to dietary guidelines and mean daily intake of food components in the diet among GI cancer survivors and the reference population; all analyses were adjusted for lifestyle and stratified for time since diagnosis and tumor location. A better understanding of the diet quality among GI cancer survivors may help us to decide whether these patients are in need of targeted interventions aimed at their diet.

## 2. Materials and Methods

### 2.1. Design

This study was conducted within the framework of the population-based Lifelines study. Lifelines is a cohort study which aims to identify the genetic, environmental, and micro-environmental factors, and their interactions associated with healthy aging and the development of chronic diseases. The cohort consists of 167,729 individuals living across the northern part of the Netherlands with a planned follow-up period of up to 30 years [13]. Participants were enrolled between 2006 and 2009. For the current analysis, we used the baseline assessment for the participants. At baseline, participants were requested to fill in a structured and validated self-administered questionnaire for measuring the following: demographics, health status, lifestyle, and psychosocial features. Anthropometric measurements were done at one of the Lifelines research sites. Lifelines is conducted according to the principles of the Declaration of Helsinki and approved by the medical ethics committee of the University Medical Center Groningen, the Netherlands. Written informed consent was obtained from all participants. Lifelines, as an extensive collaboration with OncoLifeS initiative, was approved by the medical ethics committee of the UMCG (No. 2010/109) and is ISO certified (9001:2008 Healthcare). It was registered in the Dutch Trial Register under the number: NL7839.

### 2.2. Study Population

The process of population inclusion for the analysis presented herein is depicted in Figure 1. The following participants were included: (1) those aged ≥18 of Dutch nationality; (2) those with complete information on the questions regarding their history of cancer (yes/no), and if yes: on the type of cancer and the time since the diagnosis of cancer; and (3) those with complete data on the dietary questionnaire. GI cancer survivors were defined as participants aged ≥40 who confirmed that they had at some point been diagnosed with digestive system cancers (including; bowel, gastric, and oesophageal cancers). The reference population consisted of those without any history of cancer. Participants in the reference population were randomly selected (in a ratio of 1/10) according to sex and age using the frequency matching method based on age groups of 40–49, 50–59, 60–69, and ≥70.

### 2.3. Food Intake Assessment

Dietary intake in Lifelines was assessed via a 110-item self-administered, semi-quantitative food frequency questionnaire (FFQ) covering the preceding month, which was designed based on a national survey on food composition in 1997 [14]. The dietary items included in the Lifelines FFQ covered ~80% variation of food and energy intake among the Dutch population. Replies for the frequency of food intake assessment were classified from “not this month” to “6–7 days a week”. In order to estimate portion size, fixed portion sizes and household measures were applied. Food groups and alcohol intake were extracted from FFQs using the Dutch food composition database [15]. Average individual food group intake was calculated by multiplying the frequency of intake of a given consumed food by its food group content, and then summing food group values across all consumed foods.

#### Diet Quality Assessment

The diet quality assessment was performed using the following methods: (i) the diet quality index was assessed according to the Lifelines diet (LLD) score using data from the FFQ [8]. The LLD score is based on the Dutch Dietary guideline from 2015, which is a compilation of current international scientific evidence relating foods and dietary habits with the development of the 10 main diet-related chronic diseases in the Netherlands, including colorectal cancer [16]. To calculate an LLD score, the items in the FFQ were classified into 22 food groups, of which nine food groups are supposed to have desired health effects (including vegetables, fruit, whole grain products, legumes and nuts, fish, oils and soft margarines, unsweetened dairy, coffee and tea), and three food groups are supposed to have undesired health effects (including red and processed meat, butter and hard margarines, and sugar-sweetened beverages). The maximum and minimum intake (in grams/1000 kcal) per food group was categorized into quintiles. The total score is the sum of the scores across all food groups with a range from 0 to 46, with a higher score representing adherence to a better diet quality. The details of the methods of developing a LLD score is explained in [8]; (ii) the level of adherence to dietary guidelines was quantified by the mean intake of food components divided by recommended intake in Dutch dietary guidelines multiplied by 100; (iii) to illustrate the daily intake of food components of diet quality index among the study population, the mean daily intake of food components, classified based on their desired and undesired health effects according to Dutch dietary guidelines, was demonstrated for the study population.

A high diet quality was defined as the score falling into the fourth and fifth quintiles of overall LLD score calculated for Lifelines participants (LLD score ≥ 25) [8]. The sufficiency in daily intake of the LLD score food components was assessed by recommended daily intake per each food group (g/day), as defined in Dutch food-based dietary guidelines [16]. To characterize diet quality among GI cancer survivors with different times since diagnosis, GI cancer survivors were classified into three groups: ≤4 years, 5–9 years time since diagnosis, and ≥10 years based on the time since diagnosis of GI cancers in order to assess diet quality in a short, medium, and long time since diagnosis. On the basis of tumor site, GI cancer survivors were defined as colorectal cancer (CRC) and other GI cancer survivors, including gastric and esophageal cancers.

### 2.4. Covariates

The following demographics and lifestyle factors were considered: age, sex, comorbid disease, educational level, smoking, alcohol consumption, body mass index (BMI), physical activity, and sedentary behavior. Demographics and lifestyle factors were defined as in previous studies [7,15]. The history of comorbidities was defined as having a history of any of the following: chronic bowel disease, inflammatory bowel disease, Crohn’s disease and polyps, chronic liver disease, cirrhosis, fatty liver, gallbladder stone, cholecystitis, diabetes, hypertension, dyslipidemia, arteritis, osteoporosis, and thyroid disease. Education level was classified into low (defined as below high school), medium (high school to college), and high (university degree). Smoking was defined as non-smokers (not smoking within past 12 months), former smokers (smoking history of >12 months and quit >1 month prior to study recruitment), and current smokers (currently smoking, or any history of smoking one month prior to study recruitment). Alcohol intake was calculated in grams per day using the Lifelines FFQ and classified into <10 g/day and ≥10 g/day. BMI was classified into three categories of <25, 25 ≤ BMI < 30 and ≥30 (kg/m^2^). Using the Short Questionnaire to Assess Health-enhancing physical activity (SQUASH), physical activities were assessed in terms of daily activities, such as leisure, household, work, school, and moderate-to-vigorous activities. Physical activity was put in two categories of <150 min/week and ≥150 min/week. Sedentary behavior was defined as ≥2 h/day TV watching.

### 2.5. Data Analysis

We conducted a post hoc power analysis given the number of available cases = 307, α = 0.05, probability of high diet quality in the control group = 0.25, and ratio of case to referent = 1/10. This study has the power of 94.96% to detect minimum odds of 1.60 for having a high diet quality among GI cancer survivors compared to the reference population.

Differences in mean LLD score were described according to demographics and health-related characteristics among GI cancer survivors and the reference population. First, the normal distribution of the LLD score was confirmed using PP plots, then analysis of variance (ANOVA) was applied to compare the mean LLD score across subgroups of characteristics. In the next step, the association between being a GI cancer survivor (yes/no) and high diet quality (LLD score ≥ 25) was assessed. The analyses were conducted for overall GI cancer survivors and GI cancer survivors stratified by time since diagnosis (≤4, 5–9, and ≥10 years) and tumor site (CRC and other GI cancers). For this, logistic regression analyses were performed, and odds ratios (ORs), as well as 95% confidence intervals (95% CIs) were estimated. Having a high diet quality was the outcome, and adjusted for potential confounders. These variables were selected among the baseline characteristics, using an evaluation test, if the estimated odds in terms of comparing the study groups were modified by ≥10% after including each of these variables in the analysis. Adherence to dietary guidelines in GI cancer survivors and the reference population for food components of the LLD score was quantified by the mean intake of each food components in the study groups divided by the recommended intakes in the Dutch dietary guidelines [16] and multiplied by 100. Mean daily intake of food components was compared among GI cancer survivors and the reference population using multivariate logistic regression, after being adjusted for the aforementioned covariates. In the post hoc analyses, the mean LLD score was compared according to the age and sex category among GI cancer survivors stratified by time since diagnosis and tumor site. All statistics were calculated with SPSS, version 23 (SPSS Inc., Chicago, IL, USA).

## 3. Results

Table 1 summarizes the demographics and lifestyle factors of the study population in relation to the LLD score in the reference population and GI cancer survivors. Among the reference population, LLD scores were statistically significantly higher in those aged ≥55, female gender, those with a high educational level, those who were overweight, and those who demonstrated a high level of physical activity and a lower level of sedentary behavior (*p* < 0.001). Among GI cancer survivors, the mean diet quality was significantly lower among those with age < 55, male gender, lower educational level, and high alcohol intake (*p* < 0.01, Table 1).

GI cancer survivors had similar odds for having a high diet quality (LLD score ≥ 25) compared to the reference population (OR = 0.97, 95% CI: 0.73 to 1.23). GI cancer survivors stratified for time since diagnosis had no significant association with a high diet quality compared to the reference group (OR = 1.12, (95% CI: 0.77–1.61) for ≤4 years, OR = 0.81, 95% CI: 0.52–1.26 for 5 to 9 years, and OR = 0.93, 95% CI: 0.60–1.46 for ≥10 years of time since diagnosis). Stratification for tumor sites yielded a non-significant association between a high diet quality and GI cancer survivors (OR = 1.05, 95% CI: 0.80–1.38 for CRC and OR = 0.71, 95% CI: 0.42–1.18 for other GI tract cancer survivors) (Table 2).

The intake of vegetables, unsweetened dairies, and nuts and legumes were almost 50% lower than the dietary guideline’s recommended amounts in GI cancer survivors and in reference population (Figure 2). The mean daily intake of undesired food components, including red and processed meat, butter and hard margarine, sugar-sweetened beverages, and alcohol intake was at least one serving a day among GI cancer survivors and the reference population (Appendix A). Further analyses demonstrated among GI cancer survivors, younger individuals and men 5 to 9 years after the diagnosis of cancer, as well as those with a tumour site other than colorectal had lower diet quality (Appendix A).

## 4. Discussion

In this cross-sectional study which evaluated the diet quality among GI cancer survivors in comparison with a reference population who had no history of cancer, diet scores in GI cancer survivors were not different from the reference population. Stratification for time since diagnosis and tumor site gave similar results. The intake of vegetables, unsweetened dairies, and nuts and legumes in GI cancer survivors as well as in the reference population were almost 50% lower than the dietary guideline’s recommended amounts. Daily intake of food components with undesired health effects was almost one serving a day among the study population.

The finding that the diet quality among GI cancer survivors was similar to the age- and sex-matched reference population is controversial. A study from the UK reported findings regarding bladder cancer survivors also indicating a similar diet quality among survivors of this cancer type compared to the general population of the same age [17]. However, other studies from the US reported a lower diet quality among cancer survivors compared to reference individuals [18,19]. The observed controversy can be explained in part by the fact that in the latter studies [18,19], diet quality was investigated among survivors of all cancer types without stratification for different cancer types, and without adjustment for age; whereas in our study, we did adjust for age.

We observed a higher intake of some healthy food groups, such as fish and whole grain products, among GI cancer survivors. However, the dietary recommendations for other healthy food were not met for GI cancer survivors, nor for the reference population. This was also reported in another study where dietary recommendations were not met for specific nutrients and food groups, including whole grains and vitamin D, in survivors of any cancer type [17]. On the other hand, the mean intake of sugar-sweetened beverages, hard margarine, and red and processed meat was at least one serving/day among GI cancer survivors, while these food components are advised against by the dietary guidelines [16]. Our findings were consistent with those of Zhang et al., who reported a high intake of empty calories, saturated fatty acids, and poor micronutrient intake among cancer survivors of all cancer types [18]. The undesired diet quality among GI cancer survivors might due to frequently presented symptoms among GI cancer survivors. It is likely that proper nutritional interventions, along with medical interventions, may ease clinical symptoms improving the quality of life in cancer survivors.

The current findings regarding a low intake of foods with a desired impact on health among GI cancer survivors, with gastrointestinal discomfort as a common symptom presented among GI cancer survivors and the reference population, demonstrate an unbalanced diet in both groups, highlighting the need for nutritional intervention for the whole population. Investigations indicate that an unbalanced diet is associated with a 10% increased risk of chronic disease [20]. Since morbidities among cancer survivors are higher compared to the healthy population, it is necessary to investigate the association of an unbalanced diet with increased morbidity and mortality in GI cancer survivors. The latter population has been scarcely assessed, and findings from a recent investigation indicate that a higher intake of plant-based foods and a lower intake of animal products before and after CRC diagnosis was associated with longer survival [20]. Moreover, findings from a population-based study with 13.5 years of follow-up demonstrate that a higher adherence to Dutch dietary guidelines is associated with lower risk of mortality from chronic pulmonary obstructive disease, CRC, and depression [21].

With regard to diet quality, in terms of demographics and lifestyle factors, we found a poor diet quality among younger GI cancer survivors who were men, as well as those with a lower educational level, and those with high alcohol consumption. These findings are in line with those of previous studies that report a lower diet quality in male cancer survivors [22,23]. We did not find any difference in diet quality among subgroups of other health-related factors, including smoking, which is not in line with the findings of Zhang et al., who reported a poor diet quality among cancer survivors who were current smokers in the USA [18]. The observed inconsistency can be explained by differences in the study populations from various nations with different cancer types.

It is worth noting that improving the diet quality by implementing dietary interventions may lead to a longer healthy life in cancer survivors. To achieve this goal, it is crucial to characterize the people in greater need of dietary interventions. The observed low intake of healthy foods and high intake of unhealthy foods among GI cancer survivors might demonstrate the potential relationship between frequently presented symptoms among GI cancer survivors, affecting GI tract function and food ingestion. These gastrointestinal discomfort symptoms include the existence of severe chronic pains, a low level of energy, altered sense of food taste, as well as difficulties in food ingestion in esophagus and gastric cancer survivors [24]. It is therefore essential that a team of physicians and nutritionists collaborate on palliative care and providing enteral nutritional supplements based on individual requirements to improve diet quality in affected patients.

## 5. Study Limitations and Strengths

This large population-based study, including 10% of the inhabitants of the north of the Netherlands, has several strong points, including the fact that it is a large population-based study with accurate data collection methods. However, to define GI cancer survivorship, we used self-reported data, which can be considered as a limitation. Nevertheless, the possible misclassification was minimized using Dutch cancer registry data for validating self-reported GI cancer rates. As a result of the cross-sectional design of the study, it was not feasible to assess persistent changes in diet quality after cancer detection, though by assessing the diet quality among cancer survivors with different times since cancer diagnosis, we compensated for this limitation. Moreover, the current investigation is among the first studies to assess the diet quality of GI cancer survivors using a food-based diet quality index which is based on the latest evidence-based dietary recommendation, which takes into account evidence on diet and CRC risk. Considering the obstacles related to accurate nutrient intake assessments due to measurement errors in food questionnaires [25], using food-based dietary guidelines for assessing diet quality is considered to be more applicable in routine clinical practice.

## 6. Conclusions

Diet quality in GI cancer survivors was similar to the reference population; it was found that both groups needed to improve their diet quality. Well-targeted dietary interventions are required to improve the diet quality among GI cancer survivors, as well as the general population. Further investigations are required to assess the effect of proper dietary interventions on GI cancer survivorship.

## Figures and Tables

**Figure 1 nutrients-12-02232-f001:**
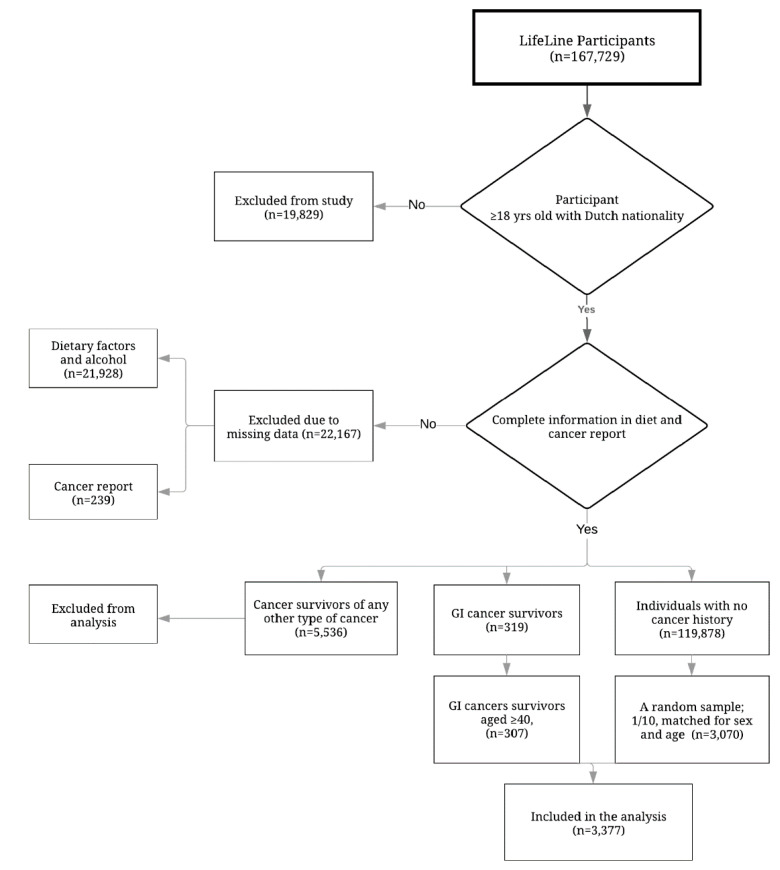
Study flow chart.

**Figure 2 nutrients-12-02232-f002:**
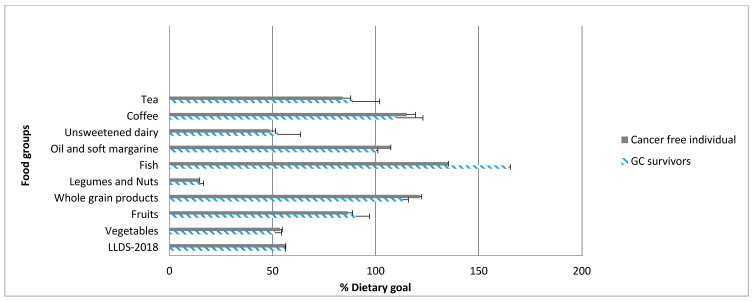
Adherence to dietary guidelines in terms of intake of foods with desired health effects in GI cancer survivors and the reference population in the Lifelines study, 2006–2016. The lengths of bars for each food component correspond to the mean intake divided by the recommended intake multiplied by 100. Recommended intake is based on the Dutch dietary guidelines published by Kromhout et al., 2015. Nutritional goals are set at 100 when the mean intake reaches the recommended intake. Abbreviation: GI, gastrointestinal.

**Table 1 nutrients-12-02232-t001:** Differences in mean Lifelines Diet (LLD) score according to demographics and health-related characteristics of the study population stratified by cancer status; Lifelines (2006–2009) ^1^.

Characteristics ^2^	Reference Population (*n* = 3070)		GI Cancer Survivors (*n* = 307)	
	*n* (%)	LLD Score Mean (SE)	*n* (%)	LLD Score Mean (SE)
Age at inclusion (years)				
<55	878 (28.60)	24.31 (0.25)	76 (24.80)	24.39 (0.68)
≥55	2192 (71.40)	26.56 (0.10)	231 (75.20)	26.15 (0.38)
*p*-value		<0.001		<0.01
Sex				
Men	1560 (50.80)	24.75 (0.14)	155 (50.80)	24.85 (0.44)
Women	1510 (49.20)	27.13 (0.15)	151 (49.20)	26.60 (0.50)
*p*-value		<0.001		<0.01
Comorbid chronic disease ^3^				
No	875 (28.50)	25.92 (0.20)	77 (25.10)	25.79 (0.65)
Yes	2195 (71.50)	25.92 (0.12)	230 (74.90)	25.69 (0.39)
*p*-value		0.98		>0.05
Educational level				
Low	1472 (48.50)	25.80 (0.15)	162 (53.10)	25.09 (0.47)
Medium	829 (27.30)	25.37 (0.21)	61 (20.00)	25.80 (0.71)
High	736 (24.20)	26.72 (0.21)	85 (26.90)	27.10 (0.63)
*p*-value		<0.001		<0.05
Smoking				
Never	1115 (35.90)	26.32 (0.17)	91 (29.80)	25.80 (0.63)
Former	1510 (50.20)	26.37 (0.14)	175 (57.40)	25.82 (0.44)
Current	426 (13.90)	23.31 (0.30)	39 (12.80)	24.97 (0.98)
*p*-value		<0.001		>0.05
Alcohol consumption				
<10 g/day	789 (25.70)	26.04 (0.21)	90 (23.90)	27.06 (0.62)
≥10 g/da	2281 (74.30)	25.88 (0.12)	217 (70.70)	25.15 (0.39)
*p*-value		0.06		<0.01
Categorized BMI (kg/m^2^)				
<25	1028 (35.40)	25.60 (0.18)	104 (13.70)	26.24 (0.57)
25 ≤ BMI < 30	1486 (48.50)	26.45 (0.14)	139 (36.80)	25.67 (0.48)
≥30	553 (18.00)	25.42 (0.26)	64 (49.50)	24.95 (0.79)
*p*-value		<0.001		>0.05
Physical activity (min/week)				
<150	1259 (41.00)	24.85 (0.17)	124 (40.40)	25.13 (0.55)
≥150	1811 (59.00)	26.66 (0.13)	183 (59.60)	26.10 (0.42)
*p*-value		<0.001		>0.05
Sedentary Behavior ^4^				
No	2452 (79.90)	25.70 (0.19)	60 (19.50)	26.53 (0.70)
Yes	618 (20.10)	26.76 (0.23)	247 (80.50)	25.51 (0.39)
*p*-value		<0.001		>0.05

^1^ ANOVA test was applied to compare mean of LLD score. ^2^ Missing values (*n*, %): Education (35, 1), Smoking (21, 0.6), BMI (3, 0.1). ^3^ Defined as history of any of the following chronic bowel diseases: inflammatory bowel disease, Crohn’s disease and polyps, chronic liver disease, cirrhosis, fatty liver, gallbladder stone, cholecystitis, diabetes, hypertension, dyslipidemia, arteritis, osteoporosis, and thyroid disease. ^4^ Defined as ≥2 h/day TV watching. Abbreviations: BMI, body mass index; GI, gastrointestinal; LLD, Lifelines diet; SE, standard error.

**Table 2 nutrients-12-02232-t002:** Association between gastrointestinal (GI) cancer survivors and high diet quality ^1^, unadjusted and adjusted analysis, stratified by time since diagnosis and tumor site.

Study Population	Individuals with High Diet Quality (%)	Unadjusted Analysis	Adjusted Analysis
		OR ^2^ (95% CI)	OR (95% CI)
Reference population	60.10	ref	ref
GI cancer survivors	59.60	0.98 (0.77 to 1.24)	0.97 (0.73 to 1.23)
Reference population	60.10	ref	ref
GI survivors (stratified by time since diagnosis)			
≤4 years	61.40	1.05 (0.74 to 1.50)	1.12 (0.77 to 1.61)
5–9 years	55.20	0.82 (0.53 to 1.25)	0.81 (0.52 to 1.26)
≥10 years	61.40	1.05 (0.68 to 1.62)	0.93 (0.60 to 1.46)
Reference population	60.10	ref	ref
GI cancer survivors (stratified by tumor site)			
CRC	58.30	1.04 (0.80 to 1.36)	1.05 (0.80 to 1.38)
Other GI tract cancer	61.50	0.80 (0.48 to 1.29)	0.71 (0.42 to 1.18)

^1^ Defined as the score falling into fourth and fifth quintiles of overall LLD score among lifelines study participants (LLDS ≥ 25). ^2^ OR (95% CI) was derived from logistic regression model; in the adjusted analyses, the following variables were included in the model: age (continuous), sex, education level (low, medium, high), smoking status (never, former, current), BMI (<25 kg/m^2^, 25 to <30 kg/m^2^ and ≥30 kg/m^2^), and physical activity (<150 min/week. and ≥150 min/week). Abbreviations: CI, confidence interval; CRC, colorectal cancer; GI cancers, gastrointestinal cancer; GI, gastrointestinal; OR, odd ratio; Ref, reference.

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
