# Peer review of "Assessment of Diet Quality and Adherence to Dietary Guidelines in Gastrointestinal Cancer Survivors: A Cross-Sectional Study"

_nutrients, 2020, doi:10.3390/nu12082232_

Round 1

Reviewer 1 Report

The manuscript (ID: nutrients-865538) is concerning on diet quality in cancer patients population. The results are not surprisingly and not add any important information that could be used for treatment, diagnosis, or prevention of GI cancers.

The authors gathered a vast amount of data and to my mind, not use them enough. They could conduct more deeply analysis, concerning on quantitive variables, f.e., examine the relation between the stage (in years) or diagnosis of cancer and physical activity/BMI/nutrients intake.

Included tables are hard to read and need to be improved.

The discussion could also focused on the potential relationship between symptoms which are the most often in patients with gastrointestinal tract cancers, their nutritional status and nutrition value of diet (according to author results).

Authors could refer to other studies examined problems with GI tract and results (obtained here) about the frequency of food intake to provide a hypothesis that this group of patients should be more educated in diet field and also be under the care of a nutrition specialist.

Specific comments:

Abstract:

- All analyses were adjusted for lifestyle. – the word „factors” should be added. Please, mention lifestyle components which were used to this analysis.

- Diet scores in GI cancer survivors were not different from the reference population (OR=0.97, 95%CI: 0.73-1.23)  - The differences in sum score of scale should be examined with using the test which compares the group (f.e. U-Mann Whitney, t-student, ANOVA).. the OR  is a measure of association between exposure and an outcome, f.e., diagnosis of cancer/health

Line 63: "Design"  - it looks like the title of the section. Please, verify

Line 71: „in participants” is it necessary in this sentence?

Line 78: Study population – it also looks like the title of the section. Please, check the correctness of the headings and all parts in the manuscript.

Line 79: according to guidelines of Journal: Abbreviations should be defined in parentheses the first time they appear in the abstract, main text, and in figure or table captions and used consistently thereafter.

In one sentence, authors used full stop after the abbreviation, and in another, they did not.

Line 80: after 1. point is colon - „:” it should be changed to other punctuation mark f.e. semicolon and after 3 is the unnecessary comma

Line 88: Power analysis should be a subsection of Statistical analysis

Line 125: Please, using abbreviations or word „years.”

Lifestyle factors: Why authors performed the analysis with the qualitative parameters only? Why did not include a quantitative measure of BMI, alcohol intake or physical activity? Analysis based on quantitative measures could be more sensitive.

Line 156: authors mentioned the included factors twice times (line 140 also). It could be omitted

Line 163-164: aforementioned – please connect

Line 171: „those with BMI ranged  25-30" can be changed to "those with overweight"

Line 174-175: Why authors calculated odd ratio? The differences in sum score of scale should be examined with using the test which compares populations (f.e. U-Mann Whitney, t-student, anova)

Line 182: "Figure 2" should be in parentheses

Line 187, 189: Table 2 and Figure S1& S2.  should be in parentheses

Table 1 and Table 2: Please, add space between "Table" and "1" or "2"

The tables required more attention and correction.

Figure 2: Only one colour is explained. (The second reflect the patients with GI cancers, I supposed)

There were no differences between these groups, so to my mind the Figure 2 is unnecessary.

Discussion section

The first study should be an introduction to discussion f.e. indicate what was the aim of the study or refer to GI cancer issue

Line 256: "Study limitation and strenght" should be bolded.

Author Response

We thank the reviewer #1 for his/her critical reviewing of our manuscript and for constructive comments. Here below we address his comments and the corresponding changes we applied to our manuscript in point by point fashion.

Comments from reviewer #1

  • The authors gathered a vast amount of data and to my mind, not use them enough. They could conduct more deeply analysis, concerning on quantities variables, f.e., examines the relation between the stage (in years) or diagnosis of cancer and physical activity/BMI/nutrients intake.

Reply: The aim of our study is to assess the diet quality status among GC survivors and compare it with a reference population; therefore, assessing the relation between the stages (in years) or diagnosis of cancer and physical activity/BMI/nutrients intake is beyond our aim. However, to properly use the available data, in line with the aim of the study, we conducted further analysis to assess the difference in mean diet quality score in adult GI cancer survivors stratified by socio-demographic characteristics and health behaviors.

Please see Page, line, ‘Mean diet quality was significantly lower among GI cancer survivors with lower education level and high alcohol consumption (Table S2)’. Diet quality was not significantly different in GI cancer survivors classified by time since diagnosis, tumour site, smoking, weight status, physical activity, sedentary behavior and comorbid disease, please see Table S2.

  • Included tables are hard to read and need to be improved.

Reply: Looking at included Tables, we assume this comment is likely meant for layout of the Tables. To improve the readability of the Tables, we edited the layout of all tables in main text. In supplementary file Table S1, we removed Coefficient and only presented the P-value for the difference in LLD score-2018 and food components among GI survivors and reference population. We added Table S2, presenting Mean LLDS-2016 score in adult GI survivors classified by socio-demographic characteristics and health behaviors.

  • The discussion could also focused on the potential relationship between symptoms which are the most often in patients with gastrointestinal tract cancers, their nutritional status and nutrition value of diet (according to author results).

Reply: We did not investigate the association between symptoms and diet. To accommodate this comment we added the following statement on page 9 lines 246 to 250 ‘The undesired diet quality among GI cancer survivors might demonstrate the potential role of frequently presented symptoms among GI cancers survivors, affecting GI tract function and food ingestion. It is likely that proper nutritional interventions a long with medical interventions may ease clinical symptoms improving the quality of life in cancer survivals.’

  • Authors could refer to other studies examined problems with GI tract and results (obtained here) about the frequency of food intake to provide a hypothesis that this group of patients should be more educated in diet field and also be under the care of a nutrition specialist.

Reply: The requested section is added to discussion on page … lines,,, and now it reads as; ‘The observed low intake of healthy foods and high intake of unhealthy foods among GI cancer survivors might demonstrate the potential relationship between frequently presented symptoms among GI cancers survivors, affecting GI tract function and food ingestion. These gastrointestinal discomfort symptoms includ existence of severe chronic pains, low level of energy, altered sense of food taste as well as difficulties in food ingestion in esophagus and gastric cancer survivors (ref). It is essential that a team of physicians and nutritionists collaborate on palliative care and providing enteral nutritional supplements based on individual requirements to improve diet quality in affected patients. ’ in Page 10 Lines278 to 285.

Abstract:

1.5          All analyses were adjusted for lifestyle. – the word „factors” should be added. Please, mention lifestyle components which were used to this analysis.

Reply: The editions in abstract is done as follows; “All analyses were adjusted for lifestyle factors’’.

The life style factors used for adjustments include: education level (low, medium high), smoking status (never, former, current), BMI (<25 kg/m2, 25 to <30 kg/m2 and ≥30 kg/m2), and physical activity (<150 mins/wk. and ≥150 mins/wk.). Due to word limitation in abstract section, we did not mention the included factors, however, in manuscript this has been mentioned in methods section on pages. 3&4lines 127 to 144 .and Table 2.

  • Diet scores in GI cancer survivors were not different from the reference population (OR=0.97, 95%CI: 0.73-1.23) The differences in sum score of scale should be examined with using the test which compares the group (f.e. U-Mann Whitney, t-student, ANOVA). the OR  is a measure of association between exposure and an outcome, f.e., diagnosis of cancer/health

Reply: To assess association between being a GI cancer survivor and having a high diet quality (here considered as outcome), compared to reference population, We applied Anova to assess the difference between mean of diet score among GI cancer survivors stratified by socio-demographic characteristics and health behavior, We next applied logistic regression model both in an un-adjusted analysis and adjusted analysis for demographics and health related characteristics. When, the unadjusted analysis using linear regression is Anova test, the adjusted analysis provides further information on the effect of covariables which might affect the observed association. s. We presented these results on pages 7, Table 2 and supplementary Table S2.

  • Line 63: "Design" - it looks like the title of the section. Please, verify

Reply: We corrected all titles and subtitles.

  • Line 71: „in participants” is it necessary in this sentence?

Reply: We dropped the sentence.

  • Line 78: Study population – it also looks like the title of the section. Please, check the correctness of the headings and all parts in the manuscript.

Reply: We edited all headers and sub headers in the text.

  • Line 79: according to guidelines of Journal: Abbreviations should be defined in parentheses the first time they appear in the abstract, main text, and in figure or table captions and used consistently thereafter. In one sentence, authors used full stop after the abbreviation, and in another, they did not.

Reply: We corrected this and now the text is consistent with abbreviations

  • Line 80: after 1. point is colon - „:” it should be changed to other punctuation mark f.e. semicolon and after 3 is the unnecessary comma

Reply: The edited is done in the text.

  • Line 88: Power analysis should be a subsection of Statistical analysis

Reply: Power analysis in now moved to data analysis section, please check

  • Line 125: Please, use abbreviations or word „years.”

Reply: We applied this suggestion and used the abbreviation form of years, in entire text of manuscript.

  • Lifestyle factors: Why authors performed the analysis with the qualitative parameters only? Why did not include a quantitative measure of BMI, alcohol intake or physical activity? Analysis based on quantitative measures could be more sensitive.

Reply: In attempt to adjust for BMI, alcohol intake or physical activity, we included continuous variables first, no significant change ware observed between adjusted and unadjusted analysis, later given the low clinical relevance of one point increment in adjusting variables in classified analysis, we used categorical variables in exploring the associations, adjustments and assessing the difference in diet quality among GI cancer survivors classified for different categories of these factors.

  • Line 156: authors mentioned the included factors twice times (line 140 also). It could be omitted

Reply: Thanks for your remark, the requested edition is applied.

  • Line 163-164: aforementioned – please connect

Reply: The required edition is done

  • Line 171: „those with BMI ranged 25-30" can be changed to "those with overweight"

Reply: Te required edition is done

  • Line 174-175: Why authors calculated odd ratio? The differences in sum score of scale should be examined with using the test which compares populations (f.e. U-Mann Whitney, t-student, anova)

Reply: We applied logistic regression model to assess the difference between GC survivors and reference population with regard to having a high diet quality, both in an un-adjusted analysis and adjusted analysis for demographics and health related characteristics. The unadjusted analysis provides same results with ANOVA and adjusted analysis provides us with further information on the adjusting variables affecting the observed association.

1.20        Line 182: "Figure 2" should be in parentheses, Line 187, 189: Table 2 and Figure S1& S2.  should be in parentheses

Reply: This is corrected.

1.22.       Table 1 and Table 2: Please, add space between "Table" and "1" or "2"

Reply: This is corrected,

1.23.       The tables required more attention and correction.

Reply: We applied required editions and corrections in Tables.

1.24.       Figure 2: Only one colour is explained. (The second reflect the patients with GI cancers, I supposed)

Reply: Figure 2 is corrected now, both colors are explained.

1.25.       There were no differences between these groups, so to my mind the Figure 2 is unnecessary.

Reply: We admit that there is no difference both in mean intake and meeting the dietary recommendations in both groups. However, Figure 2 provides us with crucial information on the level of adherence to dietary recommendations for intake of foods with desired health effects in study population, which in turn leads to clinical insights in conducting proper nutritional interventions.

Discussion section

1.26.       The first study should be an introduction to discussion f.e. indicate what was the aim of the study or refer to GI cancer issue

Reply: We added a short introduction to discussion; ‘In this cross sectional study we evaluated the diet quality among GI cancer survivors in comparison with a reference population with no history of cancer. Diet scores in GI cancer survivors were not different from a reference population. Stratification for time since diagnosis and tumour site gave similar results. The intake of vegetables, unsweetened dairies, nuts and legumes in GI cancers survivors as well as in the reference population were almost 50% below the dietary guidelines recommended amounts. Daily intake of food components with undesired health effects were almost one serving a day among the study population’. Page 9, lines 222 to 223.

1.27.       Line 256: "Study limitation and strength" should be bolded

Reply: We edited accordingly..

Reviewer 2 Report

The manuscript nutrients-865538 by Sara Moazzen et al, entitled “Assessment of diet quality and adherence to dietary guidelines in gastrointestinal cancer survivors” is an interesting epidemiologic work describing diet quality in short- and long-term Gastrointestinal (GI) cancers survivors (307) compared to a cohort of reference population (3070). The manuscript represents the first study assessing diet quality of GI cancer survivors using a food-based diet quality index which is based on dietary recommendation taking evidence on diet and CRC risk into account. Nevertheless, the manuscript requires major intervention, particularly in the language that is complex and, in some parts, incomprehensible.

The Introduction is short, well designed, but should be improved and the discussion is complete, mostly in the description of the study’s limitations and strengths.

However, the results are very poor and requires some additions. For example, the authors could include the data shown in the Supplementary file into the Results Section and describe them.

I have, moreover, some minor adjustments:

  • In the sentence: “GI cancers survivors were defined as participants aged ≥40 who confirmed that they had ever been diagnosed with digestive system cancers (including; bowel, gastric and oesophageal cancers)”, after the word “including”, please, correct with colon; also in the sentence: “including; being”.
  • Table 1: lacks titles in the two different columns.
  • Figure 2: the sided titles are incomplete as well as the legend.
  • Figure S1: “Time sine diagnosis” in the first histogram, please, correct.

Author Response

We appreciate very much reviewer number 2 for reviewing and commenting on our manuscript.  Here follow we address his comments and highlight the corresponding changes we made on our manuscript.

Comments from reviewer #2

The manuscript nutrients-865538 by Sara Moazzen et al, entitled “Assessment of diet quality and adherence to dietary guidelines in gastrointestinal cancer survivors” is an interesting epidemiologic work describing diet quality in short- and long-term Gastrointestinal (GI) cancers survivors (307) compared to a cohort of reference population (3070). The manuscript represents the first study assessing diet quality of GI cancer survivors using a food-based diet quality index which is based on dietary recommendation taking evidence on diet and CRC risk into account.

2.1.         Nevertheless, the manuscript requires major intervention, particularly in the language that is complex and, in some parts, incomprehensible.

Reply: A comprehensive language edition is now done in entire text of manuscript by language editor.

2.2.         The Introduction is short, well designed, but should be improved and the discussion is complete, mostly in the description of the study’s limitations and strengths.

Reply: We assume the reviewer refers to “improving” the language quality of introduction.  We applied a through language edition in introduction.

2.3.         However, the results are very poor and require some additions. For example, the authors could include the data shown in the Supplementary file into the Results Section and describe them.

Reply: We believe that including supplementary data into main text, will turn it to a lengthy results section. It will detract the readers and overdue the word limits. We improved result section by providing further supplementary analysis on diet quality among GI cancer survivors classified by demographics and health characteristics, please see Table S2 and result section which now reads as; ‘Mean daily intake of undesired food components including red and process meet, butter and hard margarine, sugar sweetened beverages and alcohol intake was at least one serving a day among GI cancer survivors and the reference population (Table S1). Mean diet quality was significantly lower among GI cancer survivors with lower education level and high alcohol consumption (Table S2).Further analyses demonstrated among GI cancer survivor, younger individuals and men with 5 to 9 years after the diagnosis of cancer, and those with tumour site other that colorectal had lower diet quality, (Figure S1& S2). ‘ in page 4&5, line 185 to 191.

2.4.         In the sentence: “GI cancers survivors were defined as participants aged ≥40 who confirmed that they had ever been diagnosed with digestive system cancers (including; bowel, gastric and esophageal cancers)”, after the word “including”, please, correct with colon; also in the sentence: “including; being”.

Reply: The required edition is done.

2.5.         Table 1: lacks titles in the two different columns.

Reply: We thoroughly checked all the tables and applied necessary corrections.

2.6.         Figure 2: the sided titles are incomplete as well as the legend.

Reply: The sided title is now complete, and we made the required editions in legend.

2.7.         Figure S1: “Time sine diagnosis” in the first histogram, please, correct.

Reply: Thanks for your remark, We applied the required correction

Reviewer 3 Report

Authors have attempted to evaluate for differences in diet quality among the normal population and GI cancer survivors. They have chosen the reference normal population from the same database - Lifelines from 2006 until 2009. Using this arm as a comparator arm for analysis gives the results only at "that" time of analysis. This could merely be used as a comparison between two different populations at that time - baseline characteristics.

However, since the effects of diet on tumorigenesis are temporal in nature, lack of complete follow up (30 years as mentioned by authors) is a major limitation. Ideally, the dataset at the end of the complete follow-up (ongoing questionnaires) would provide meaningful answers by exempting people who developed GI cancers during the follow-up period and should have used that population as the reference arm. In my opinion, I feel that the authors shouldn't have chosen baseline questionnaires as their data points which might not provide a realistic picture and they should have used an on-going questionnaire for analysis, which will provide a better evaluation of dietary patterns between two groups. 

Author Response

We highly appreciate reviewer number 3 for reviewing and commenting on our manuscript.  Here follow we address his/her comments and highlight the due to changes we made on our manuscript.

Comments from reviewer #3

Authors have attempted to evaluate for differences in diet quality among the normal.GI cancer survivors. They have chosen the reference normal population from the same database - Lifelines from 2006 until 2009. Using this arm as a comparator arm for analysis gives the results only at "that" time of analysis. This could merely be used as a comparison between two different populations at that time - baseline characteristics.

3.1.         However, since the effects of diet on tumorigenesis are temporal in nature, lack of complete follow up (30 years as mentioned by authors) is a major limitation. Ideally, the dataset at the end of the complete follow-up (ongoing questionnaires) would provide meaningful answers by exempting people who developed GI cancers during the follow-up period and should have used that population as the reference arm. In my opinion, I feel that the authors shouldn't have chosen baseline questionnaires as their data points which might not provide a realistic picture and they should have used an on-going questionnaire for analysis, which will provide a better evaluation of dietary patterns between two groups. 

Reply: We do admit the limitation of our study, given the cross sectional design and we have highlighted this in study limitations section, page 9, lines 288 to 290; “As a result of the cross-sectional design of the study, it was not feasible to assess persistent changes in diet quality after cancer detection, though by assessing the diet quality among cancer survivors with different time since cancer diagnosis, we compensated for this limitation.” We found the comparable mean diet quality among GI cancer survivors with >4 years, 4-9years and over 10 years . time since diagnosis (P=0.15, Please see Table S2). This finding suggests the consistency of the findings on diet quality among GI cancer survivors.

Further, it is worthwhile to note that dietary assessments by regionally validated FFQ with 110 food items could be a proper representative of main dietary habits of a study population which will remain rather constant over time.

Round 2

Reviewer 1 Report

The detailed list of authors replies is very satisfactory, and the revision has much improved the manuscript.

Some very minor changes should be made:

Line 191: lower diet quality,
A coma at the end of the sentence is unnecessary.

Figure 1: Could the authors edited the flow chart and did more carefully some elements of them, f.e. arrows and properly text-align „Included the analysis.”

Author Response

We appreciate the reviewers for providing us with further revisions/feedbacks. Here below, we address their comment and the corresponding corrections we applied to our manuscript.

Comment from Reviewer #1

The detailed list of authors replies is very satisfactory, and the revision has much improved the manuscript.

Some very minor changes should be made: Line 191: lower diet quality,
A coma at the end of the sentence is unnecessary.

Figure 1: Could the authors edited the flow chart and did more carefully some elements of them, f.e. arrows and properly text-align „Included the analysis.”

Reply: We made the required edition in line 191. We edited figure 1, adjusted the alignments and edited the sentence; ‘Included in the analysis’.

Reviewer 2 Report

In this revised form, the manuscript could be accepted for the publication in Nutrients.

Author Response

We appreciate the reviewers for providing us with further revisions/feedbacks. Here below, we address their comment and the corresponding corrections we applied to our manuscript.

Comment from reviewer #2

In this revised form, the manuscript could be accepted for the publication in Nutrients.

Reply: We appreciate the feedback from the respected reviewer.

Reviewer 3 Report

The authors' rebuttal and correction are noted. No other suggestions at this time. With the acknowledgement of limitations of the study, this manuscript provides important information regarding diet quality and adherence in GI malignancy survivors.  

Author Response

We appreciate the reviewers for providing us with further revisions/feedbacks. Here below, we address their comment and the corresponding corrections we applied to our manuscript.

Comments from reviewer #3

The authors rebuttal and correction are noted. No other suggestions at this time. With the acknowledgement of limitations of the study, this manuscript provides important information regarding diet quality and adherence in GI malignancy survivors.

Reply: We thank the respected reviewer for positive feedback.
